# Ladostigil Attenuates Induced Oxidative Stress in Human Neuroblast-like SH-SY5Y Cells

**DOI:** 10.3390/biomedicines9091251

**Published:** 2021-09-17

**Authors:** Keren Zohar, Elyad Lezmi, Tsiona Eliyahu, Michal Linial

**Affiliations:** 1Department of Biological Chemistry, Institute of Life Sciences, The Hebrew University of Jerusalem, Jerusalem 91904, Israel; keren.zohar@mail.huji.ac.il (K.Z.); tsiona.e@mail.huji.ac.il (T.E.); 2Department of Genetics, Institute of Life Sciences, The Hebrew University of Jerusalem, Jerusalem 91904, Israel; elyad.lezmi@mail.huji.ac.il

**Keywords:** Alzheimer’s disease, neurodegenerative disease, mitophagy, microglial activation, RNA-seq, UPR, ER stress

## Abstract

A hallmark of the aging brain is the robust inflammation mediated by microglial activation. Pathophysiology of common neurodegenerative diseases involves oxidative stress and neuroinflammation. Chronic treatment of aging rats by ladostigil, a compound with antioxidant and anti-inflammatory function, prevented microglial activation and learning deficits. In this study, we further investigate the effect of ladostigil on undifferentiated SH-SY5Y cells. We show that SH-SY5Y cells exposed to acute (by H_2_O_2_) or chronic oxidative stress (by Sin1, 3-morpholinosydnonimine) induced apoptotic cell death. However, in the presence of ladostigil, the decline in cell viability and the increase of oxidative levels were partially reversed. RNA-seq analysis showed that prolonged oxidation by Sin1 resulted in a simultaneous reduction of the expression level of endoplasmic reticulum (ER) genes that participate in proteostasis. By comparing the differential gene expression profile of Sin1 treated cells to cells incubated with ladostigil before being exposed to Sin1, we observed an over-expression of Clk1 (Cdc2-like kinase 1) which was implicated in psychophysiological stress in mice and Alzheimer’s disease. Ladostigil also suppressed the expression of Ccpg1 (Cell cycle progression 1) and Synj1 (Synaptojanin 1) that are involved in ER-autophagy and endocytic pathways. We postulate that ladostigil alleviated cell damage induced by oxidation. Therefore, under conditions of chronic stress that are observed in the aging brain, ladostigil may block oxidative stress processes and consequently reduce neurotoxicity.

## 1. Introduction

A decline in cognitive function is a hallmark of normal brain aging. Morphological changes and alterations in cell-type composition are often observed in the aged brain [1]. The deficit in learning and memory shows a decreased number of neurons, reduction of synaptic sites, and changes in the properties of dendritic spines [2,3]. On a cellular level, a decline in the brain’s higher functions is associated with apoptotic death [4], reduced mitochondrial function [5], proteotoxicity [6,7], and enhanced autophagy [8]) among other outcomes. A characteristic phenomenon of the aging brain cells is the reduced capacity to cope with oxidative stress compared to cells from young-adult brains [9]. A failure in antioxidant defense mechanisms in the aging brain [10,11] and impaired mitochondrial function [12] are associated with memory decline [13]. Once the cellular compensatory mechanisms can no longer cope with the stress [14], pathological processes are induced [15,16] as observed following brain injury [17] and in neurodegenerative diseases.

A single molecule of ladostigil can inhibit the enzymatic activity of monoamine oxidase and cholinesterase. In addition, ladostigil enhances neurogenesis and acts as an antidepressant. Ladostigil reduces TNF-α in lipopolysaccharide (LPS)-activated microglia and in the spleen of LPS-injected mice [18]. Aged rats that were treated with ladostigil for a long period (6 months) confirmed the drug’s anti-inflammatory activity [19,20]. Such treatment partially blocked the age-dependent decline in learning and memory [19,21,22]. In addition, ladostigil treatment changed the degree of microglial activation in aging rats [23]. In the young rats, most microglia are at their resting state but become activated along with aging. The effect of chronic treatment with ladostigil on microglia morphology in old rats is consistent with the anti-inflammatory role in the brain. Ladostigil treatment affected the profile of gene expression in the aging brain. Many of these affected genes act in neurogenesis, synapse formation, calcium, and fatty-acid homeostasis [22].

Towards an understanding of the ladostigil mode of action, in vitro experiments that mimic brain inflammation were conducted. Activating primary neonatal microglia with lipopolysaccharides (LPS) and ATP [24] resulted in TNFα, IL-1β, and IL-6 secretion [25]. These pro-inflammatory cytokines contribute to learning deficits and loss of synapses [26]. In vitro studies showed that ladostigil suppressed TNFα and IL-1β release, presumably by a reduction in crucial phosphorylation sites that drive Nf-κB translocation to the nucleus [20]. Nf-κB is a bridge between pro-inflammatory cytokine expression and neurogenesis [27,28] and an attractive therapeutic route for neurodegenerative diseases [29].

One of the outcomes of microglial activation (as well as that of macrophages, neutrophils, and endothelial cells) is the accumulation of reactive oxygen species (ROS) which include superoxide, singlet oxygen, hydroxyl radical, and hydrogen peroxide (H_2_O_2_) [30]. These molecules can damage proteins, DNA, and lipids at the cell membranes, ultimately leading to pathological changes of cell structure and function [31]. Mitochondria are the primary source of ROS in cells. Upon oxidative stress condition, apoptosis and mitophagy are activated for attenuating any potential damage [32,33]. In the mammalian brain, neurons are strongly dependent on oxidative metabolism for ATP production, and therefore, are prone to ROS-dependent damage. An imbalance between ROS production and antioxidants occurs in Alzheimer’s disease, Parkinson’s disease, and other types of dementia of the elderly [34].

In this study, we tested the effect of ladostigil on a homogenous culture of neuroblastoma-derived SH-SY5Y cells, a derivative of the human SK-N-SH cells. Undifferentiated SH-SY5Y that contain voltage-dependent calcium channels [35] and active large dense cored vesicles are similar to sympathetic neurons [36]. We used these cells to assess their oxidative vulnerability [37]. We tested SH-SY5Y cells as a homogenous cellular setting. The action of ladostigil was assessed by cell survival assay, redox measurements, and differential gene expression (RNA-seq technology) in naïve and under induced stress conditions. Analysis of cells exposed to oxidative stress conditions indicated down-regulated expression of genes that function in folding and proteostasis. We show that in the presence of ladostigil, genes related to membrane dynamics are affected. We discuss the capacity of ladostigil to cope with accumulated stress, given its beneficial effects in reducing inflammation and neurotoxicity.

## 2. Materials and Methods

### 2.1. Materials

MTT (3-(4,5-dimethylthiazol-2-yl)-2,5-diphenyltetrazolium bromide) was used for cell viability. All reagents were purchased from Sigma-Aldrich (Burlington, MA, USA) unless otherwise stated. Ladostigil, (6-(N- ethyl, N- methyl) carbamyloxy)-N propargyl-1(R)-aminoindan tartrate, was a gift from Spero BioPharma (Tel Aviv, Israel). Media products MEM and F12 (ratio 1:1), heat-inactivated fetal calf serum (10% FCS), L-Alanyl-L Glutamine, and all tissue culture materials were purchased from Biological Industries (Kibbutz Beit-Haemek, Israel). H_2_O_2_ (Sigma-Aldrich; Burlington, MA, USA) was diluted in cold water and the solutions were used within 2 h. Sin1 (3-(4-5 Morpholinyl) sydnone imine hydrochloride (Cat-M5793; Sigma-Aldrich, Burlington, MA, USA). Lipofectamine-2000 (Thermo-Fisher, Waltham, MA, USA) was used as a transfection reagent according to the manufacturer’s instructions.

### 2.2. SH-SY5Y Cell Culture

Human neuroblastoma SH-SY5Y cells were purchased from ATCC (American Type Culture Collection, Rockville, MD, USA) [38]. Cells were cultured in Minimum Essential Media (MEM and F12 ratio 1:1, 4.5 g/L glucose) with 10% fetal calf serum (FCS) and 1:10 L-Alanyl-L-Glutamine. Cell cultures were incubated at 37 °C in a humidified atmosphere of 5% CO2. A protocol in which ladostigil and the subjected stressors were incubated together failed to monitor a robust biological effect (data not shown). Thus, ladostigil was added to the culture medium 2 h prior to the activation of oxidative stress to cells. Cells were tested 24 h later (i.e., 26 h after the addition of ladostigil) unless otherwise stated.

### 2.3. Cell Viability Assay

SH-SY5Y cells were cultured at a density of 2 × 10^4^ cells per well in 96-well plates, in 200 µL of the medium. MTT assay used colorimetric measurement for cell viability, where viable cells produced a dark blue formazan product. Cells were treated in the absence or presence of ladostigil, at different concentrations. The drug was solubilized in water. MTT solution in phosphate-buffered saline (PBS, pH 7.2) was prepared at a working stock of 5 mg/mL. After 24 h, the culture medium was supplemented with 10 µL of concentrated MTT per well. Absorption was determined in an ELISA-reader at λ = 535 nm, using a reference at 635 nm. Cell viability was expressed as a percentage of the viability of untreated cells. Each experimental condition was repeated 8 times.

### 2.4. Oxidative Stress Sensor

Cells were transfected with molecular sensors that allow a change in fluorescence signal normalized by GFP [39]. Cyto-roGFP plasmids and Mito-matrix-roGFP are redox-sensitive biosensors where the oxidative state of cysteine affects the intensity of fluorescence. The plasmids were a gift from the laboratory of D. Reichmann (The Hebrew University of Jerusalem) and are identical to the commercial Addgene vectors (Watertown, MA, USA) [40]. SH-SY5Y cells (CRL-2266) were cultured in 6- or 12-well plates at 37 °C and 5% CO_2_, to reach a 70–80% confluent level (2 × 10^4^ per cm^2^). For transfection, the roGFP plasmids (Cyto and Mito-roGFP at a 1:1 ratio) were added at room temperature to the culture in MEM and F12 (at a cell density of 2 × 10^4^ per cm^2^). A total of 5.5 μg plasmid DNA was mixed with transfection reagent in 6-well cell cultures and incubated for 24–36 h before analysis by flow cytometry (FACS). The 405-nm and 488-nm lasers were used for excitation, and the matched 525 nm and 530-nm optical filters were used for fluorescence detection, respectively. We measured the ratio between 488 nm (reduced) to 405-nm (oxidized). Dithiothreitol (DTT) was used for the calibration of maximal reduction, while diamide was used to achieve maximal cells oxidation. The percentage of dead cells was quantified by Propidium Iodide (PI). Appendix A shows the results of DTT and diamine calibration along with the naive untreated cells’ redox state.

### 2.5. Reverse Transcription Polymerase—Chain Reaction (RT-PCR)

Total RNA was extracted with TRIzol (Thermo-Fisher Scientific, Waltham, MA, USA), and RT was performed using a Ready-To-Go first-strand synthesis kit (Cytiva, Marlborough, MA, USA) according to the manufacturer’s instructions. RNA was reverse transcribed into cDNA (1 μg) and used in the PCR reaction. The PCR conditions consisted of denaturation at 95 °C for 2 min and 35 cycles (10 s at 95 °C, 15 s at 60 °C, 30 s at 72 °C), and 5 min for a final extension. The PCR products were separated on 1.5–2% agarose gel and stained with ethidium bromide, followed by densitometry measurement (using image processing ImageJ program from GitHub). In addition, TaqMan real-time PCR was performed according to the indicated exon-boundary designed primers (Thermo-Fisher Scientific, Waltham, MA, USA). Table 1 lists the forward and reverse primers (F and R, respectively) used for the gel-based RT-PCR and primers for the real-time qPCR reaction. β-actin was used for normalization and was included in the RT-PCR reaction mix (for distinctive amplicon sizes).

### 2.6. RNA Sequencing

SH-SY5Y cells were pre-incubated for 2 h with ladostigil at a concentration of 5.4 μM. Cells were exposed to Sin1 (*t* = 0) and harvested at 10 h or 24 h following the addition of ladostigil. Total RNA was extracted using the RNeasy Plus Universal Mini Kit (QIAGEN, Hilden, Germany) according to the manufacturer’s protocol. Total RNA samples (1 μg RNA) were enriched for mRNAs by pull-down of poly(A), and libraries were prepared using the KAPA Stranded mRNA-Seq Kit according to the manufacturer’s protocol. RNA-seq libraries were sequenced using Illumina NextSeq 500 to generate 85 bp single-end reads.

### 2.7. Differential Expression Analysis

Next-generation sequencing (NGS) data underwent quality control using FastQC, aligned to the reference genome GRCh38 with STAR aligner [41] using default parameters. Genomic loci were annotated using GENCODE version 37 [42]. Following normalizations, low expressing genes below a threshold of 5 counts-per-million (CPM) in three samples were filtered out. Genes with a false discovery rate (FDR) adjusted *p*-value ≤ 0.05 and an absolute log fold-change (FC) above 0.3 were considered as significantly differentially expressed (unless otherwise mentioned).

### 2.8. Statistics

All experiments were performed with a minimum of three biological replicates. Principal component analysis was performed using the R-base function “prcomp”. The counts per gene were normalized using the “weighted” Trimmed Mean of M-values (TMM) approach, according to the Bioconductor package EdgeR [43]. We report here on genes that met the FDR adjusted *p*-value of ≤0.05 (q-value). Enrichment analyses for pathways used GO pathway based on slim GO biological processes (PANTHER ver 16) [44]). All figures were generated using the ggplot2 R package.

### 2.9. Data Availability

RNA-seq data files from SH-SY5Y experiments were deposited to ArrayExpress under the accession E-MTAB-10817.

## 3. Results

### 3.1. SH-SY5Y Cell Survival upon Acute Oxidative Stress

A recent study compared rat brain transcriptomes from young and old rats (6.5 and 22 months, respectively) in brain regions that participate in information integration and memory consolidation [22]. Chronic treatment with ladostigil in rats (from 16 to 22 months) resulted in improved performance in learning and memory tasks. Treatment with ladostigil attenuated the activation of microglia in the aged brain [22,23]. Therefore, we ask whether ladostigil can dissipate induced stress at a cellular level. To this end, we used the neuroblastoma-derived SH-SY5Y cells as a homogenous cellular setting. To test the sensitivity of SH-SY5Y cells to a short-lived oxidative insult, we exposed cells to hydrogen peroxide (H_2_O_2_) and measured cell viability 24 h later using MTT assay (Figure 1). Cells that were exposed to a moderate level of H_2_O_2_ (up to 80 μM) remained viable, suggesting that SH-SY5Y cells can dissipate a moderate level of H_2_O_2_-induced oxidative stress (Figure 1A).

To test a possible shift in the cellular redox state in the presence of H_2_O_2_, we utilized molecular sensors for oxidation [45]. Cells were transfected with a mixture of cytosolic and mitochondrial fluorescence sensors for monitoring their redox state (see Materials and Methods, Appendix A). In cells exposed to H_2_O_2_ (80 μM) for 3 h, a marked increase in the fraction of oxidized cells was observed (28%, *p*-value <0.001, Figure 1B). We found that pre-incubation with ladostigil (2 h, 5.4 μM) significantly reduced the oxidative state compared to unexposed cells (to 87%, *p*-value <0.05) (Figure 2B). Ladostigil showed no effect on cell survival at varying concentrations (5.4 μM and 54 μM; Appendix A). We concluded that SH-SY5Y cells cope successfully with acute oxidative stress, and therefore H_2_O_2_ might be a less appropriate stressor for measuring the long-term effects of stress on neuronal-like cells.

### 3.2. SH-SYS5 Redox State under Induced Long-Term Oxidation Stress

We replaced H_2_O_2_ with Sin1 (3-morpholinosydnonimine), a reagent that promotes prolonged oxidative stress [46] that damages brain mitochondria [47]. Sin1 is a long-lasting stressor that simultaneously generates nitric oxide (NO) and superoxide (O2−), and acts as a peroxynitrite donor. We calibrated the effect of a mild yet long-term oxidative stress by Sin1 as a substitute for chronic stress occurring along with the brain’s aging. Figure 2A shows the results of cell viability assays in the presence of Sin1. For untreated cells, an apparent increase in cell proliferation was observed in the presence of ladostigil. However, cell viability remains stable in the presence of Sin1 at 50 μM and 100 μM. In contrast, a strong reduction in cell viability was recorded at higher Sin1 concentrations (300 μM; 18% survival). Under such conditions, pre-incubation of ladostigil (2 h) improved viability to a level of ~75% relative to untreated cells. Cells’ protective effect by ladostigil was not observed at a higher Sin1 concentration (500 μM, Figure 2A).

We tested whether the effect of Sin1 can be directly monitored by the cell’s redox state, using FACS-based measurements (Appendix A). Figure 2B shows that at moderate levels of Sin1 (100 μM), the fraction of oxidized cells was increased after 5 h (by ~20%), and an elevated oxidative state lasted for 24 h. Under a high Sin1 concentration (300 μM), the fraction of the oxidized cell was further elevated (from 14% to 37% for 100 μM and 300 μM of Sin1, respectively). Note that the level of cell oxidation was maximal at 5 h following the exposure of Sin1 at 300 μM (Figure 2B).

We questioned whether the physiological effect of Sin1 (with and without ladostigil) is exposed by the gene products that control the cellular redox. We compared the expression of enzymes that protect cells against apoptotic death [48] by catalyzing oxidation-related species. For example, Sod genes catalyze the dismutation of superoxide radical anion to H_2_O_2_ and free oxygen.

Figure 3A shows results of RT-PCR for the ROS-sensitive genes with Sod1, Sod2, and Gpx1 (Glutathione peroxidase 1). In the presence of Sin1, the expression levels of Sod2, a mitochondrial resident enzyme, and Gpx1 were upregulated, but down-regulated to the baseline level by a high concentration of ladostigil (Figure 3A). Figure 3B shows the results of quantitative real-time PCR (qPCR) for several redox-sensitive genes. Ladostigil reduced the expression of the genes encoding the antioxidant superoxide dismutase enzymes (Sod1, Sod2) and Gpx1 by ~50–60% of the maximal induced level by Sin1. The same trend was observed for Nfe2l2, an oxidation-sensitive gene regulatory transcription factor (Nrf2). We also tested Mif (Macrophage migration inhibitory factor). Similarly, in the presence of a high concentration of ladostigil, and Sin1 induction Mif expression level was 39% of its level induced by Sin1. Mif is a proinflammatory cytokine with an oxidoreductase activity [49]. It was postulated as a chaperone, allowing protecting the mitochondria and the ER from accumulated misfolded Sod1 [50].

We concluded that ladostigil takes part in suppressing the elevation in the oxidized state in response to a mild level of Sin1.

### 3.3. A Simultaneous Suppression of ER Chaperones and Folding Enzymes by a Chronic Oxidative Stress

We performed RNA-seq (in biological triplicates) to compare the transcriptome of untreated cells and Sin1-treated cells. We tested the transcriptome 24 h after exposure to monitor the adapted cellular response. Among the genes that are robustly expressed (total 11,532 genes, see Materials and Methods), 84% were unchanged, 9% were induced, and 7% were suppressed (Appendix A).

Figure 4A shows a volcano plot comparing non-treated naïve cells and cells exposed to a mild level of Sin1 (50 μM) after 24 h. Note that most statistically significant genes are downregulated by Sin1 (colored blue). We compared the significantly down- (Figure 4B) and up-regulated (Figure 4C) genes following Sin1 treatment relative to non-treated cells and assessed their annotations’ enrichment according to GO (Gene Ontology) biological process terms. The most enriched terms for the suppressed genes (*n* = 402) include different aspects of response to unfolding proteins, de novo folding, and response to ER stress [51]. Testing the enrichment for the induced genes (*n* = 493) resulted in lower enrichment relative to the suppressed genes (Figure 4B,C, *x*-axis) with dominant GO biological process terms concerning neuronal development and differentiation. We concluded that Sin1 treatment led to changes in the protein folding response and ER stress, while the induced genes highlight neuronal differentiation.

Figure 5 summarizes all 35 strongly down-regulated genes (>2 folds; minimal *p*-value FDR <1 × 10^−12^). The gene list is enriched with chaperones, enzymes that form multiprotein complexes of the endoplasmic reticulum (ER), and protein folding regulators. According to the cellular pathways map (ENRICHR [52]), Dnajc3, Xbp1, Hspa5, Dnajb11, Hyou1, Calr, Hspa1b, Herpud1, Pdia4, Hsp90b1, and Hspa1a are associated with protein processing and ER quality control, showing enrichment of 40.9 folds (*p*-value FDR of 2.5 × 10^−13^). Major chaperones (Hspa5, Hspa1b, Hspa1a, Hsp90b1, and Hyou1) that belong to the Hsp90 and Hsp70 families are located at the ER lumen. In addition, among the genes suppressed by Sin1 are co-chaperones of BiP (Dnajc3 and Dnajb11 of the Hsp40 family). These genes are required for proper folding, trafficking, or degradation of proteins. For example, when unfolded (or misfolded) proteins in the ER exceed the capacity of the folding machinery, a redundant set of heat shock proteins (Hsp) initiates the unfolded protein response (UPR). Among the listed genes, Hyou1 carries signals that connect the ER and mitochondria, the known Ca^2+^ storage organelles. In this line, calreticulin (Calr), an ER-resident protein, is a Ca^2+^-binding chaperone that acts in neuronal regeneration capacity following injury [53]. Another ER-resident protein is Manf (Mesencephalic astrocyte-derived neurotrophic factor). It is an ER-stress regulated protein that exhibits cytoprotective by regulating ER homeostasis [54]. We concluded that suppression of the ER maintenance machinery reflects the initial elevation in cell oxidation that potentially suppresses reduced ER protein production in SH-SY5Y cells. We postulate that following the stress induced by Sin1, suppression in the ER misfolding response reflects cells’ adaptation (Figure 5).

Table 2 lists the genes which expression is significantly induced (>2-folds). A comparison of over-expressed versus down-regulated genes showed that their absolute expression levels were significantly lower (TMM of 11.7 versus 78.8, respectively; *p*-value = 0.019). Pdk1 (pyruvate dehydrogenase kinase 1; 2.23-fold, *p*-value = 8.5 × 10^−92^) is among the few highly induced genes. Overexpression of Pdk1 in animal models and cell lines shows some resistance to Aβ peptides and other neurotoxins [55]. Several of the induced genes are associated with neuronal pathology. For example, Erich3 (2.5-fold, *p*-value = 7.6 × 10^−49^) plays a role in vesicular trafficking and neurotransmitter actions, and a change in its expression occurs after antidepressant treatment [56].

About a third of the significantly induced genes (6 out of 19 genes, >2 fold) are non-coding RNAs (ncRNAs, Table 2). The ncRNA Rn7sl1 (RNA component of signal recognition particle 7SL1) is part of a riboprotein complex that acts in ER translocation. Another notable gene is P2rx7 (Purinergic receptor P2x7; 1.99-fold, *p*-value FDR = 2.8 × 10^−28^), an ATP receptor which activation leads to change in Ca^2+^ homeostasis and an ER-dependent Ca^2+^ cytotoxicity [57]. A full list of induced genes (*n* = 493) is available in Appendix A.

### 3.4. Ladostigil Alters the Expression of Genes Act in Dissipating Oxidative Damage

Incubation of SH-SY5Y cells with ladostigil had a negligible effect on basal gene expression. RNA-seq analysis for ladostigil treated relative to untreated cells showed only 7 (total 11,588; Appendix A) genes were statistically significant differentially expressed). Among the down-regulated genes is Tmem87a, an endosome-to-TGN retrograde transport (*p*-value FDR = 0.0054) that activates anterograde transport of GPI-anchored and transmembrane proteins. Tmem129 (induced by 24%, a *p*-value FDR = 0.025) is an ER-resident of protein that is part of the dislocation complex. The abundant mitochondrial NADH-dehydrogenase 6 (Mt-Nd6) [58] was suppressed by 21%. Oxidative stress led to a diffusion of the Mt-Nd6 RNA from its matrix location [59]. We concluded that ladostigil per se has a negligible effect on SH-SY5Y cells at a resting state. However, the nature of affected genes suggests an involvement of the mitochondria and ER.

Figure 6 shows the differential expression of genes in cells exposed to Sin1 and pretreated with ladostigil. Appendix A displays the differential expressed ratio (log FC) as a function of the raw expression data (logCPM, *x*-axis). Altogether, only 10 genes (TMM >5; *p*-value FDR <0.05) show a significant differential expression when compared to Sin1 treated cells (Appendix A). The most significantly upregulated gene is Clk1 (Cdc-like kinase 1). Clk1 is a multifunctional kinase that phosphorylates serine/arginine (SR) proteins, which affect mRNA splicing. Clk1 was identified as a gene that shows a reduced amount in the hippocampus following chronic stress, while treatment with clomipramine, an antidepressant drug, prevented the stress effect on its expression [60]. As Clk1 was also associated with the pathophysiology of Alzheimer’s disease, it was declared a promising therapeutic target [61]. Appendix A lists the differentially expressed genes along with their statistical significance.

Synaptojanin 1 (Synj1, *p*-value FDR = 2.1 × 10^−5^) is among the very few genes downregulated by ladostigil. The role of Synj1 is to facilitate the recycling of synaptic vesicles in neurons. Synj1 role in regulating autophagy was reported based on Synj1-deficient mice that were associated with an impairment in age-dependent autophagy in multiple brain regions [62]. Moreover, the reduction of Synj1 accelerated the clearance of the toxic Aβ peptide and consequently attenuated cognitive decline [63]. The other ladostigil suppressed gene is Ccpg1 (cell-cycle progression gene 1), an ER-resident protein that participates in autophagy of the ER (denoted ER-phagy) and membrane dynamics. In numerous cells, the Ccpg1 gene was induced by the UPR, thus linking ER stress with ER-phagy [64]. Whether in Sin1 stressed cells, ladostigil interferes with the UPR response (e.g., Xbp1), and ER-phagy is yet to be investigated.

## 4. Discussion

A significant reduction in the age-dependent decline of spatial memory and learning was observed following a long-term treatment of rats by ladostigil (1 mg/kg/day; for 6 months). The anti-inflammatory activity mediated by microglia underlies the neuroprotective action of ladostigil [20,23]. Results from RNA-seq analysis of aged rat brains revealed substantial alterations in gene expression as a consequence of long-term ladostigil treatment [22]. However, the complexity of the cell composition in the brain (e.g., microglia, neurons, astrocytes) masked cell-specific effects.

Cells that are strongly dependent on mitochondrial function (e.g., neurons, muscle) and protein secretion (e.g., neurons, microglia, endocrine cells) are suitable to investigate stress-dependent mechanisms. In this study, we used undifferentiated SH-SY5Y as a cellular model for studying chronic stress underlying CNS pathology. The observed sensitivity of SH-SY5Y cells to changes in their redox level (Figure 1, Figure 2 and Figure 3) and ER proteostasis (Figure 4 and Figure 5) is intriguing in view of numerous neuronal and chronic diseases that are characterized by a strong interplay between oxidative and ER stress [65].

The pathology of brain aging is associated with an imbalance between the antioxidant response mechanism and internal ROS generation [66]. In this study, we tested the impact on gene expression 24 h after stress induction. We showed that under acute exposure to H_2_O_2_, SH-SY5Y cells have a substantial capacity to overcome the damage (Figure 1). We showed that under a high concentration of oxygen peroxide ladostigil fails to rescue cells from apoptotic death (Figure 1). Notably, in cultured cells, H_2_O_2_ can only act for a few minutes [67]. In general, excess oxidants may compromise cell function by modifying proteins, lipids, and cell energetics, ultimately leading to cell death. Our results emphasize the difference between acute, short-lived stress by H_2_O_2_ and a long-lived stressor of Sin1 (Figure 2). Sin1 is a source of reactive peroxide that causes lipid peroxidation and mitochondrial dysfunction, and its impact on cells lasts for hours [68,69]. We showed that priming cells by ladostigil before Sin1 enables cells to preferably cope with stress (Figure 2 and Figure 3).

Down-regulation of genes involved in ER, chaperons, and quality control is a characteristic of Sin1-exposed cells (50 µM, 24 h). Inspection of the expression profile of these Sin1 exposures revealed a link between the redox state and protein folding homeostasis (Figure 4). At the organism level, failure in maintaining a healthy redox state triggers pathological conditions that eventually lead to diseases (e.g., neurodegenerative, cardiovascular, metabolic disorders). In the aging brain, it is mostly the astrocytes and activated microglia that secrete extracellular cytokines that lead to accelerated ER malfunction. Under mild ER stress, the UPR system is turned on, in parallel to the shut-down of protein production [70]. In SH-SY5Y cells, ladostigil altered a handful of genes that act in the ER (Figure 6). We propose that protein overload (due to a suppressed capacity to resolve misfolding and aggregates) can lead to mitochondrial function disturbance. Autophagy (ER-phagy and Mitophagy) is a mode in which cells cope with the toxic accumulation of ROS and nitroperoxides affecting organelle integrity [64].

In summary, we characterized SH-SY5Y cells as an attractive model for exposing the impact of mild but chronic stress. We argue that the molecular response is likely to be cell-type specific. Thus, cells with low antioxidative activity are prone to damage by external insults [71]. Neurons are exceptionally sensitive to nitrogen and oxygen-based damage due to their dependence on the respiratory chain [72]. Upon differentiation of SH-SY5Y (e.g., by retinoic acid, neurotrophic factor), the cells undergo morphological changes and start expressing numerous neuronal markers. Differentiated SH-SY5Y cells are often used for studying CNS neuron properties (e.g., dopaminergic neurons, glutamate to GABA conversion, neuronal-like vesicle secretion) [72]. This system might be useful to assess the impact of ladostigil as the vulnerability of differentiated SH-SY5Y to stress is enhanced [37,73].

Under chronic oxidative stress, the redox homeostasis is disturbed, and consequently, protein folding and ER function become fragile. In SH-SY5Y cells, ladostigil affected selected genes that participate in ER-autophagy and endocytosis (Figure 6). This study characterizes the response to stress and the adaptation of SH-SY5Y cells in the presence of ladostigil. It is a step forward in deciphering the cellular and molecular response to ladostigil. However, generalizing the observations from in vitro studies to aged brains is not always verifiable. An attractive possibility is that the beneficial effect of ladostigil treatment on memory decline of aged rats is mediated by regulating oxidative stress and maintaining ER physiology of CNS neurons.

## Figures and Tables

**Figure 1 biomedicines-09-01251-f001:**
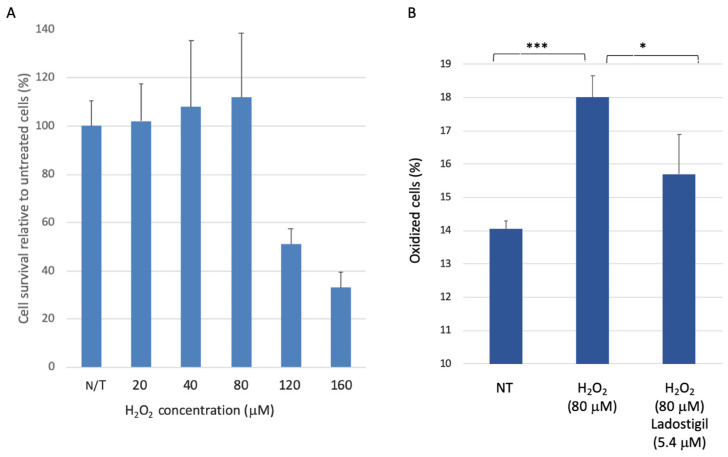
SH-SY5Y cells under acute oxidation stress and ladostigil treatment. (**A**) Cell viability with increasing concentrations of hydrogen peroxide (H_2_O_2_) was analyzed after 24 h by MTT assay. (**B**) Cells transfected with a mix of plasmids carrying sensors for redox state (see Materials and Methods) were analyzed by FACS. The cells were exposed to H_2_O_2_ for 3 h with or without pre-incubation (for 2 h) with ladostigil. In each experiment, the fraction of oxidized cells is determined relative to the untreated cells. Each group represents biological triplicates. The statistical test (*t*-test) showed significance at the level of <0.05 (*) and <0.001 (***).

**Figure 2 biomedicines-09-01251-f002:**
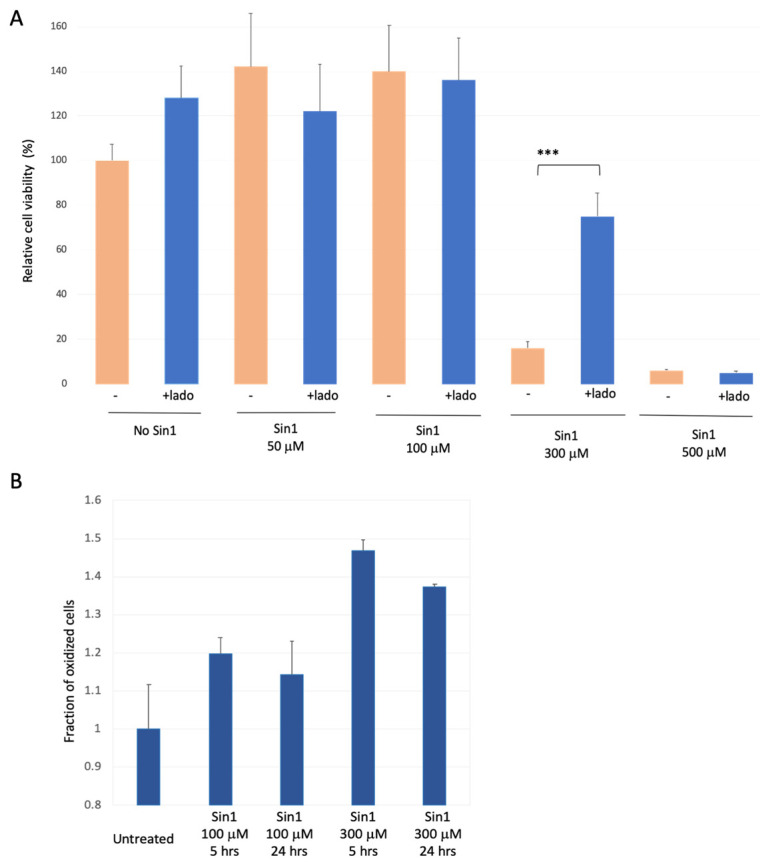
SH-SY5Y cells under chronic long-term oxidation stress and ladostigil treatment. (**A**) Cell viability upon increasing Sin1 concentration in the presence or absence of ladostigil (lado, 5.4 μM) was monitored_._ Cell survival difference that is statistically significant is marked (***, *p*-value < 0.005). (**B**) Cells were exposed to Sin1 (100–300 μM) for 5 and 24 h, and the redox state of transfected cells expressing GFP was monitored (see Materials and Methods). The fraction of the oxidized cells is determined relative to untreated cells. Experiments included internal controls of a fully oxidized and fully reduced condition (Appendix A).

**Figure 3 biomedicines-09-01251-f003:**
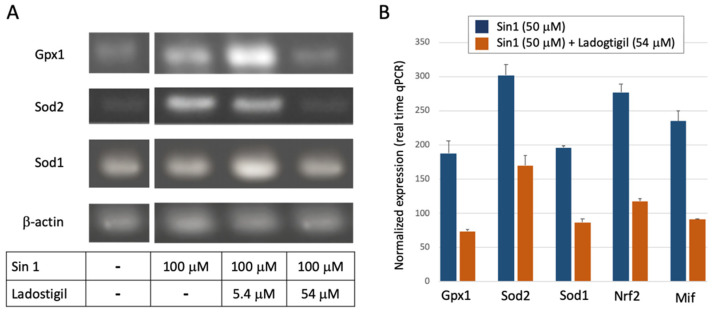
Measurement of expression for a set of ROS-sensitive genes by PCR. (**A**) Gel-based PCR amplification using primers listed in Table 1. (**B**) Results from real-time quantitative PCR (qPCR). The normalized expression of the listed genes in the presence of Sin1 (50 mM, blue) and the following preincubation with ladostigil (orange). Each gene is normalized to its basal level of untreated cells (*y*-axis, refers to 100).

**Figure 4 biomedicines-09-01251-f004:**
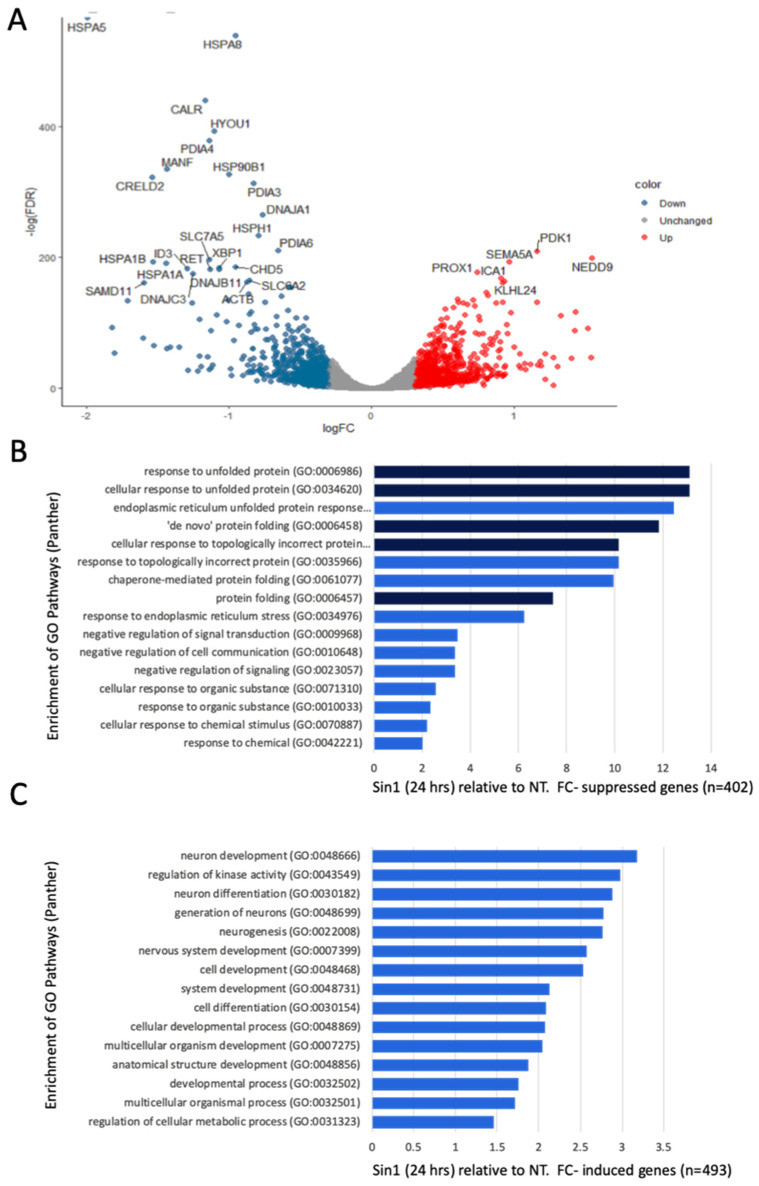
Differential expression and annotations’ enrichment comparing the expression of non-treated naïve and cells exposed to Sin1 (50 μM) for 24 h. (**A**) Volcano plot of the differentially expressed genes that were induced (red) and suppressed (blue) according to the fold change and statistical significance. (**B**) Enrichment of suppressed genes with normalized expression level >5 TMM, and <0.75-fold change. (**C**) Enrichment of induced genes with the normalized expression level of >5 TMM and >1.33-fold change. Listed are the terms with >10 genes that are statistically significant (*p*-value FDR < 0.05; blue, <0.001, dark blue). Indicated terms are from the PANTHER GO process list. FC, fold change. NT, non-treated SH-SY5Y cells.

**Figure 5 biomedicines-09-01251-f005:**
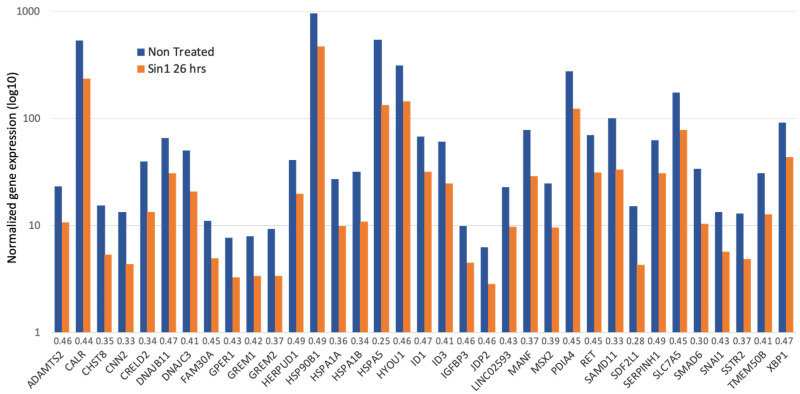
A collection of genes which expression is significantly reduced (>2 folds) following exposure to Sin1 compared to untreated cells (26 h). Gene expression levels are on a logarithmic scale. The ratio of the gene expression levels in Sin1 treated versus not treated cells is indicated next to the gene symbols. Genes are sorted alphabetically.

**Figure 6 biomedicines-09-01251-f006:**
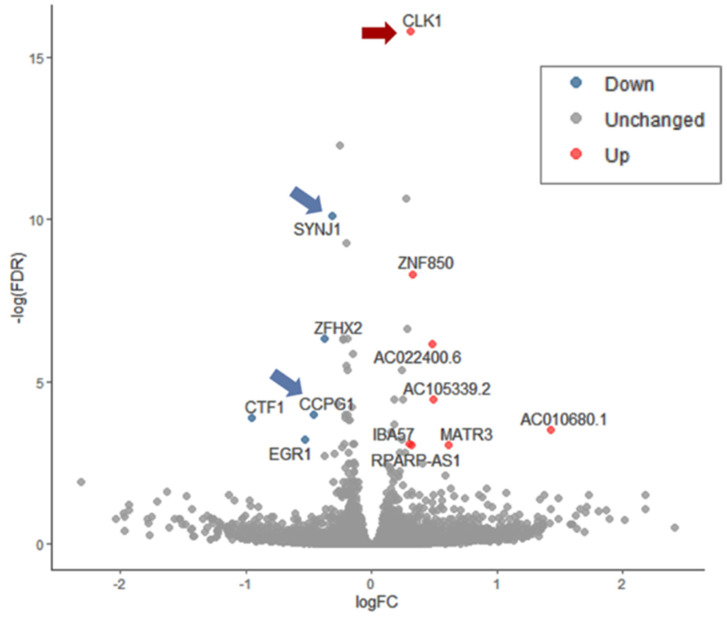
A volcano plot of differential gene expression as a result of pre-incubation (2 h) with ladostigil in the presence of Sin1 (24 h) compared to cells that were exposed to Sin1 alone. Each experimental group includes three biological replicates. RNA-seq reads were processed by a statistical model (using the edgeR, see Materials and Methods) and represented by the log FC (fold change) with FDR < 0.05. Arrows mark the statistically significant genes that function in oxidative and ER stress response.

**Table 1 biomedicines-09-01251-t001:** RT-PCR on selected genes that participate in the cell redox balance.

Gene Symbol (Synonym)	Gene Name	TaqMan ID Real Time-PCR	TaqMan Amplicon Length (nt)	Forward (F), Reverse (R) PCR Primers	PCR Amplicon Length (nt) ^a^
SOD1	Superoxide Dismutase 1	Hs00533490_m1	60	F: TTTGCGTCGTAGTCTCCTGC R: CTTTGGCCCACCGTGTTTTC	308
SOD2	Superoxide Dismutase 2	Hs00167309_m1	67	F: CTGCTCCCCGCGCTTTCTTA R: CACGTTTGATGGCTTCCAGC	373/490
GPX1	Glutathione Peroxidase 1	Hs00829989_gH	76	F: TTACAGTGCTTGTTCGGGGC R: TCTTGGCGTTCTCCTGATGC	314
NFE2L2 (NRF2)	Nuclear Factor, Erythroid 2 Like 2	Hs00975961_g1	74	F: TTATAGCGTGCAAACCTCGC R: TGTGGGCAACCTGTCTCTTCA	373
MIF	Macrophage Migration Inhibitory Factor	Hs00236988_g1	56	F: GTGGTGTCCGAGAAGTCAGG R: TTGCTGTAGGAGCGGTTCTG	324
ACTB	Actin Beta	Hs99999903_m1	171	F: ACAGAGCCTCGCCTTTGCCGA R: CATGCCCACCATCAGCCCTGG	196

^a^ The two reported sizes of the PCR products resulting from alternatively spliced variants.

**Table 2 biomedicines-09-01251-t002:** Gene expression over-expressed (>2 folds) in SH-SY5Y exposed to Sin1 (24 h) relative to untreated cells.

Symbol	Gene Name	FDR	Ratio	CPM	Label
AC006042.2	peptidylprolyl isomerase (cyclophilin)-like 4 (PPIL4) pseudogene	1.2 × 10^−6^	2.32	5.62	pseudo.
BX005019.1	novel transcript	1.8 × 10^−23^	2.05	11.43	lncRNA
ERICH3	glutamate rich 3	7.6 × 10^−49^	2.50	21.34	coding
IQCN	IQ motif containing N	1.4 × 10^−16^	2.16	10.17	coding
LINC02575	long intergenic non-protein coding RNA 2575	3.5 × 10^−39^	2.68	8.82	lncRNA
LOX	lysyl oxidase	3.3 × 10^−14^	2.06	5.36	coding
MTUS1	microtubule associated scaffold protein 1	4.5 × 10^−21^	2.41	5.83	coding
MYO15B	myosin XVB	1.7 × 10^−15^	2.47	5.26	coding
NEDD9	neural precursor cell expressed, developmentally down-regulated 9	4.0 × 10^−87^	2.91	16.74	coding
NEGR1	neuronal growth regulator 1	1.1 × 10^−57^	2.23	29.69	coding
NYAP2	neuronal tyrosine-phosphorylated phosphoinositide-3-kinase adaptor 2	2.3 × 10^−17^	2.11	6.44	coding
PDK1	pyruvate dehydrogenase kinase 1	8.5 × 10^−92^	2.23	53.06	coding
PLCL1	phospholipase C like 1 (inactive)	8.9 × 10^−16^	2.05	6.90	coding
RN7SL1	RNA component of signal recognition particle 7SL1	0.011	2.42	6.43	ncRNA
SERTM2	serine rich and transmembrane domain containing 2	6.4 × 10^−24^	2.32	6.68	coding
SLIT2	slit guidance ligand 2	3.5 × 10^−16^	2.24	5.24	coding
TSPEAR-AS1	TSPEAR antisense RNA 1	2.4 × 10^−40^	2.86	11.06	lncRNA
TSPEAR-AS2	TSPEAR antisense RNA 2	1.4 × 10^−51^	2.69	13.02	lncRNA
ZBTB20	zinc finger and BTB domain containing 20	6.3 × 10^−19^	2.27	21.96	coding

## Data Availability

The raw data is shared in ArrayExpress accession E-MTAB-10817. Normalized data is available in Appendix A and DE (differential expressed) is available in Appendix A. Normalized data is available in Appendix A and DE (differential expressed) is available in Appendix A.

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
