# Peer review of "Ladostigil Attenuates Induced Oxidative Stress in Human Neuroblast-like SH-SY5Y Cells"

_biomedicines, 2021, doi:10.3390/biomedicines9091251_

Round 1
Reviewer 1 Report
There is some good science here, but the manuscript is sloppy. The title does not match the paper. There is some data that seems to be thrown in and is not well described. There are many typos, etc. Some ideas and methods could be described better.
Review :
Ladostigil attenuates oxidation and ER stress in human neuroblast-like SH-SY5Y cells
**You don’t actually look at ER stress- the title is misleading. It would also be useful to look at another measure of redox state.
Summary:
This study was trying to follow up on rat studies that showed that a compound (ladostigil) was beneficial in neurogenerative disorders. The authors sought to study the mechanism of ladostigil in neuronal-like SH-SY5Y cells. The authors exposed the cells to acute (H2O2) and chronic (Sin1) stress measured cell viability and gene expression +/- ladostigil. They found that ladostigil altered genes in ER-autophagy and endocytic pathways. They postulated that ladostigil alleviated cell damage by oxidation and ER stress. They believe it may attenuate neurotoxicity and cell death that accompany chronic stress conditions in the aging brain.
Abstract:
Page 1 Line 11-12
This sentence is confusing Neuroinflammation resulting from the induction of oxidative stress in neurodegenerative diseases and following brain injury
Line 24-24
Therefore, it may attenuate neurotoxicity and cell death that accompany chronic stress conditions in the aging brain. – confusing
Introduction
line 37- aging brain is a reduced capacity to cope with stress (e.g., oxidative stress)- just say oxidative stress, unless there is another stress you are concerned about.
Ladostigil, is a compound with antioxidant and anti-inflammatory activity. Can you elaborate on what exactly this means? Can you comment further on its mechanism of action?
Line 44-46 You talk about the effects on microglia, but after treatment, but you don’t establish what baseline microglia function is and how that is altered in disease. Because you go on to further discuss microglia, introducing these cells properly is important.
Line 52- inconsistent capitalization of drug. This is a generic name, so it is not capitalized unless it is at the beginning of the sentence.
Line 57- needs a reference
Line 66- How are these different from neurons? Sounds like they are cancer cells, presumably so that you can culture them?
The final paragraph is very confusing as written.
Materials and Methods
Line 89- what is “Linsidomine hydrochloride Sigma Lipofectamine-2000”?
Results
Line 167
I am totally confused. I thought you were going to start with discussion of ladostigil and its effects in neuron-like cells. The first figure is in aged rat brains. The introduction and abstract should be modified accordingly. I am unclear, you may have addressed this in your materials and methods. You should clearly state your data set that you used, or you should define how you treated the rats.
Figure 1- see comments above. The figure legend says “The listed genes show a statistical significance of p-value FDR <1E-06.” I don’t see any error bars or anything in the figure to show that statistics were done. The words in the figure are fuzzy and very hard to read. Also, this would be better presented in black and white.
Why did you use the SH-SY5Y cells instead of primary cell culture? How close are they to the rat brains that you examined in figure 1?
Figure 2-
Why did you use 80 um in Figure 2B? It seemed like the cells tolerated 80 um fairly well. It would have been more useful to test a potentially “harmful” dose. What does treatment with ladostigil look like when you use 100 uM or 120 uM H2O2?
Line 222-224 “We concluded that SH-SY5Y cells cope successfully with short-term oxidation stressors and therefore are less appropriate for measuring the long-term effects of stress on neuronal-like cells.” Why? Short-term results do not always translate to long-term results.
Figure 3
Be sensitive to color blind readers with the figures
The words are fuzzy
The words are small and hard to read
Where are the statistics on B, C, and D?
Figure D- where is the GADPH control?
The labeling of the figures is really unclear.
D- It doesn’t seem like stress response genes are really changing, and one is going up?
Fig3s
This is also very confusing to read. It would be easier to see the untreated on the far left.
GPX1 just seems to be all over the place. SOD2 levels are also all over the place. I don’t know that many conclusions can be drawn from these data.
Figure 5
Please consider more disability friendly colors. Words are blurry and hard to read.
Figure 6
Consider using more conventional language like “up-regulation and down-regulation”.
Discussion
Line 463-465- This does not make sense. “n this study, we used undifferentiated SH-SY5Y cells as a model for CNS neurons. The SH-SY5Y cells exhibit neuron-like [32], primarily the interconnection between ROS (mitochondrial and cytosol) and ER proteostasis.” I still don’t really understand what SH-SY5Y cells are.
Line 486- typo
Line 503- you talked about lncRNAs earlier and now you are talking about ncRNAs.
Author Response
Review 1:
We want to thank the referee for his/her excellent comments that helped us to improve the submission substantially. Please note that the numbering in the revised manuscript was changed. The original version included seven figures and three tables. The revised version has six figures and two tables.
Ladostigil attenuates oxidation and ER stress in human neuroblast-like SH-SY5Y cells
**You don’t actually look at ER stress- the title is misleading. It would also be useful to look at another measure of redox state.
Reply: We revised the title to better reflect the presented results. The revised title is: “Ladostigil attenuates Induced oxidative stress in human neuroblast-like SH-SY5Y cells”
Summary:
This study was trying to follow up on rat studies that showed that a compound (ladostigil) was beneficial in neurogenerative disorders. The authors sought to study the mechanism of ladostigil in neuronal-like SH-SY5Y cells. The authors exposed the cells to acute (H2O2) and chronic (Sin1) stress measured cell viability and gene expression +/- ladostigil. They found that ladostigil altered genes in ER-autophagy and endocytic pathways. They postulated that ladostigil alleviated cell damage by oxidation and ER stress. They believe it may attenuate neurotoxicity and cell death that accompany chronic stress conditions in the aging brain.
Abstract:
Page 1 Line 11-12
This sentence is confusing Neuroinflammation resulting from the induction of oxidative stress in neurodegenerative diseases and following brain injury
Reply: Thanks, we revised to “Pathophysiology of common neurodegenerative diseases involves oxidative stress and neuroinflammation”.
Line 24-24
Therefore, it may attenuate neurotoxicity and cell death that accompany chronic stress conditions in the aging brain. – confusing
Reply: We simplified and replaced: Therefore, under conditions of chronic stress that signify aging brain, ladogtigil may block oxidative stress processes and consequently reduce neurotoxicity.
Introduction
line 37- aging brain is a reduced capacity to cope with stress (e.g., oxidative stress)- just say oxidative stress, unless there is another stress you are concerned about.
Reply: Replaced, as suggested.
Ladostigil, is a compound with antioxidant and anti-inflammatory activity. Can you elaborate on what exactly this means? Can you comment further on its mechanism of action?
Reply: To elaborate on the mechanism of action of ladostigil, we added explanation and evidence on aged brain and iv vitro studies (Revised lines 43-61).
Line 44-46 You talk about the effects on microglia, but after treatment, but you don’t establish what baseline microglia function is and how that is altered in disease. Because you go on to further discuss microglia, introducing these cells properly is important.
Reply: Thank you for this comment. We added a sentence on the state of microglia in young (6.5 months) and old rat brain (22 months).
Revised lines 49-53: “Microglia are the main immune effectors of the CNS, in young rat brain microglia are at their resting state but become activated upon aging. The effect of ladogtigil on microglia morphology in old rats is consistent with the anti-inflammatory role in the brain following chronic treatment.”
Line 52- inconsistent capitalization of drug. This is a generic name, so it is not capitalized unless it is at the beginning of the sentence.
Reply: Corrected.
Line 57- needs a reference
Reply: Added.
Line 66- How are these different from neurons? Sounds like they are cancer cells, presumably so that you can culture them?
Reply: While no cell-line can mimic features of primary cells, the SH-SY5Y was established as a good model for research (with many publications over the last 30 years). To avoid any confusion, we specified that we used undifferentiated cells. In the discussion, we elaborate on the benefit of studying differentiated SH-SY5Y as a more vulnerable system.
Revised lines 74-77. “Undifferentiated SH-SY5Y that contain voltage-dependent calcium channels [32] and active large dense cored vesicles are similar to sympathetic neurons [33]. We used these cells to assess their oxidative vulnerability [34].
The final paragraph is very confusing as written.
Reply: We are sorry for this unclarity. We rewrote and simplified the paragraph:
Revised lines 78-83. “The action of ladostigil was assessed by cell survival, redox measurements, and differential gene expression (RNA-seq technology) in naïve and under induced stress conditions. Analysis of cells that are exposed to oxidation stress conditions indicated the down-regulation of genes that function in folding and proteostasis. We show that in the presence of ladostigil, genes related to membrane dynamics are affected. We discuss the capacity of ladostigil to cope with an accumulated stress, with respect to its beneficial effects by attenuating inflammation and neurotoxicity in vivo”.
Materials and Methods
Line 89- what is “Linsidomine hydrochloride Sigma Lipofectamine-2000”?
Reply: Sorry, a period was missing. To clarify, we included a source and catalog number.
Revised lines 95-96: Sin1 (3-(4-5 Morpholinyl) sydnone imine hydrochloride (Sigma-Aldrich, Cat-M5793). Lipofectamine-2000 (Thermo-Fisher).
Results
Line 167
I am totally confused. I thought you were going to start with discussion of ladostigil and its effects in neuron-like cells. The first figure is in aged rat brains. The introduction and abstract should be modified accordingly. I am unclear, you may have addressed this in your materials and methods. You should clearly state your data set that you used, or you should define how you treated the rats.
Reply: The purpose of the original section 3.1 was to connect the reader with the complexity of ‘in vivo’ (i.e., old rat brain). As it led to a confusion (as you clearly stated and was also mentioned by additional referee), we remove all together this section and referred briefly to published results (Linial et al. 2020). We emphasize the rationale for the need for a simplified, and controllable in vitro cellular setting.
Revised lines 176-188.
Figure 1- see comments above. The figure legend says “The listed genes show a statistical significance of p-value FDR <1E-06.” I don’t see any error bars or anything in the figure to show that statistics were done. The words in the figure are fuzzy and very hard to read. Also, this would be better presented in black and white.
Reply: We removed Figure 1 altogether.
Why did you use the SH-SY5Y cells instead of primary cell culture? How close are they to the rat brains that you examined in figure 1?
Reply: SH-SY5Y cells are very different from (any) primary culture. Recall that maintaining primary neurons (in healthy state) often requires co-culturing (i.e. as a mixture of glia, astrocytes, and neurons or conditioned media). In short, for basic molecular and cellular functions we sought a reproduceable cell culture that can be manipulated (e.g. by transfection) and exerts homogenous characteristics for molecular profiling (e.g. RNA-seq). We clarified the rationale for using this cell line in a Revised section 3.1. Revised lines 176-188.
Figure 2-
Why did you use 80 um in Figure 2B? It seemed like the cells tolerated 80 um fairly well. It would have been more useful to test a potentially “harmful” dose. What does treatment with ladostigil look like when you use 100 uM or 120 uM H2O2?
Reply: The use of 80 mM cells for Revised Figure 1B is for monitoring acute stress (which is different from cell survival that is measured only 24 hrs after exposure). Eventually, we monitored cells in Figure 1B after 3 hrs only. We added a clarification sentence on the difference between the two methods for cell monitoring in the legend of the Figure (Revised Figure 1).
Line 222-224 “We concluded that SH-SY5Y cells cope successfully with H2O2 short-term oxidation stressors and therefore this stressor is less appropriate for measuring the long-term effects of stress on neuronal-like cells.” Why? Short-term results do not always translate to long-term results.
Reply: Indeed, short-term results do not always translate to long-term results. The signal as measured in Revised Figure 1B is a short lived. Note that after the signal was faded as shown in Supplemental Figure S2. As we aim to study the impact of ladostigil in a setting which is more appropriate to chronic conditions (as in aging, neurodegenerative diseases), we selected Sin1 as an attractive alternative.
Figure 3
Be sensitive to color blind readers with the figures
The words are fuzzy
The words are small and hard to read
Where are the statistics on B, C, and D?
Figure D- where is the GADPH control?
The labeling of the figures is really unclear.
D- It doesn’t seem like stress response genes are really changing, and one is going up?
Reply: We improved the presentation of original Figure 3 (larger fonts, simple colors). Figure2B, 2C (revised) are extracted from FACS assays. Each experiment is based on internal controls to determine the gating and to define the partition of ~50,000 cells (see Supplementary Figure S1). We included statistics for Revised Figure 2C. We removed Figure 2D to keep the figure more readable and simpler.
Fig3s
This is also very confusing to read. It would be easier to see the untreated on the far left.
GPX1 just seems to be all over the place. SOD2 levels are also all over the place. I don’t know that many conclusions can be drawn from these data.
Reply: We simplified the figure and only included the most important experimental conditions. This simplified version was added to the main text (and thus we eliminated original Figure S3). To substantiated the results in a more quantitative way, we present the real-time quantitative PCR results (Revised Figure 3B) with the statistics.
Revised lines 244-252: “Figure 3B shows the results from real-time quantitative PCR (qPCR) for several redox sensitive genes. The antioxidant superoxide dismutase enzymes Sod1, Sod2, Gpx1 and Nfe2l2, a major oxidation sensitive gene regulatory transcription factor (Nrf2) reduced their expression by ~50-60% of the maximal induced level by Sin1. We also tested Mif (Macrophage migration inhibitory factor). Similarly, in the presence of high concentration of ladostigil, and Sin1 induction, Mif expression level was 39% of its level in the presence of Sin1. Mif is a proinflammatory cytokine with an oxidoreductase activity [46]. It was postulated as a chaperone, allowing protecting the mitochondria and the ER from ac-cumulated misfolded Sod1 [47].”
Figure 5
Please consider more disability friendly colors. Words are blurry and hard to read.
Reply: We replaced as suggested. Simpler color, larger fonts.
Figure 6
Consider using more conventional language like “up-regulation and down-regulation”.
Reply: Thanks. We replaced it as proposed with this more ‘conventional’ wording throughout.
Discussion
Line 463-465- This does not make sense. “n this study, we used undifferentiated SH-SY5Y cells as a model for CNS neurons. The SH-SY5Y cells exhibit neuron-like [32], primarily the interconnection between ROS (mitochondrial and cytosol) and ER proteostasis.” I still don’t really understand what SH-SY5Y cells are.
Reply: As asked already in the introduction, we added an explanation on these cells. We rephrase this sentence that was indeed a bit confusing. Revised lines 392-396. “In this study, we used undifferentiated SH-SY5Y as a cellular model for studying chronic stress that may lead to CNS pathology. The observed sensitivity of SH-SY5Y cells to changes in their redox level (Figures 1-3) and ER proteostasis (Figures 4-5) is intriguing in view of numerous neuronal and chronic diseases that are signified by a strong interplay between oxidative and ER stress [62]”.
Line 486- typo
Reply: corrected
Line 503- you talked about lncRNAs earlier and now you are talking about ncRNAs.
Reply: We removed all together the discussion on lncRNA (or ncRNA) including the figure (asked by the second referee) and indeed it allows the presentation to remain coherent and shorter.
Reviewer 2 Report
Review of Manuscript ID: biomedicines-1360915
Ladostigil attenuates oxidation and ER stress in human neuroblast-like SH-SY5Y cells, by Keren Zohar, Elyad Lezmi, Tsiona Eliyahu and Michal Linial.
Zohar and colleagues evaluated the protective effect of ladostigil on SH-SY5Y cells exposed to Sin1-induced oxidative stress. This manuscript presents interesting data useful for deciphering the molecular effect of ladostigil in neurons. The scientific approach and hypotheses are of interest. Unfortunately, this manuscript is not mature enough yet: the overall organization of the manuscript should be improved, the interpretation of some sets of data is not correct, and the quality of some figures should be improved. Under its current form, this manuscript cannot be published in biomedicines. But provided that the main comments and questions listed below are seriously considered, this manuscript could be resubmitted as it will then be worth being published, in my opinion.
Major comments
Title
The title is not accurate as the sets of data presented in this manuscript correspond to the attenuation of the effect of oxidative stress by ladostigil. The effect of ladostigil on ER stress features have not been monitored.
Abstract
In the abstract, the authors stated that oxidative stress induced by Sin1 “resulted in coordinated suppression of ER quality control and ER stress gene sets”. The observation that the expression of genes involved in both ER quality control and ER stress is modified does not preclude these two events are coordinated.
IntroductionThe description of SHSY-5Y cells characteristics made in introduction line 67-69 (ref 31) corresponds to differentiated cells. The authors used undifferentiated cells.
Material and methods
-What is used to solubilize ladostigil?-Are untreated cells treated by ladostigil solubilization solution?-Please precise what is triggering the fluorescence change of molecular sensors cyto-roGFP and mito-matrix-roGFP, respectively.-Which SHSY-5Y cell number was used for oxidative stress experiments?-Which transfection reagent was used? A weird sentence lines 89-90 suggest it may be lipofectamine-2000?-Propidium iodide is mentioned in material and methods section but it seems it has not been used in the presented experiments.-PCR extension was really 5 minutes for 300-500 pb amplicons (line 130)?-Were real time PCRs performed with the same primers as RT-PCR described in Table 1?
Results
-Title of section 3.1 has no meaning. Moreover, as the authors only looked for immunology-related genes, they cannot conclude that an immunological signature dominates in the brain of aged rats.-The authors did not explain why they choose to specifically monitor immunology-related gene expression.-No house-keeping gene was used as control in Figure 1. It would be useful to make sure that gene expression upregulation is a general age-related effect.-There is no Y axis legend in Figure 1.-There is no standard deviation in Figure 1.-Indicate the age of the rats in Figure 1. Adult and old is not precise enough. Young rats (6.5-month-old) and old rats (22-month-old) could be clearer.-Which strain of rats was used to generate Figure 1 data? How many animals? Which sex?-It is not clear why the authors stopped using H2O2 and why they used Sin1 to induce oxidative stress instead of classical compounds like paraquat for example?-Please give more information about Sin1, like its capacity to spontaneously liberate NO when in solution.-What is the effect of ladostigil on Sin1-induced oxidative stress when cells are not preincubated with ladostigil?-The treatments’ durations are not clearly indicated on Figure 3. Cells were incubated with ladostigil for 1 or 2 hours?-On figure 3C, Sin1 treatment was 3, 5 and 8 hours as indicated on the bottom of the figure or 3 to 24 hr as indicated in the legend?-If figure 3B, 3C and 3D present data from experiments repeated 3-4 times, the standard deviations values should be indicated, together with statistical analyses.-Figure 3D should show the values of gene expression levels in the absence of Sin1 to see if Sin1 has any impact by itself. -The authors did not explain why they choose to specifically monitor Sod1, Sod2 and GPX1 genes expression.-Figure S3A interpretation is wrong. The authors probably wrongly annotated the wells.Moreover, one cannot draw any conclusion from single-PCR loaded on an agarose gel. This figure should be removed from the manuscript or this set of data should be strengthened.-The aim of the manuscript is to study the effect of ladostigil on oxidative-stressed cells. Thus Figures 4 and 5 and Table 2, in which the effect of Sin1 alone on gene expression is described, should be linked to figure 7.-How do the authors explain that no apoptosis, ER stress and UPR genes are up or downregulated upon Sin1 and Sin1+ladostigil treatments?-The authors do no explain why they were interested in lncRNA expression. Moreover, I don’t see the point of showing the effect of Sin1 alone on lncRNA expression in this manuscript. What would have been interesting is the effect of ladostigil+Sin1 on lncRNA expression.The sentence line 517 is wrong as ladostigil effect on lncRNA expression level has not been studied here.-What is the effect of ladostigil itself on genes and lncRNA expression levels?-Title of section 3.6 has no meaning.
-Table S1 and Table S2 are missing in the supplementary file.
-What is the difference between Figure 7 and Figure S4 besides their formats?
Minor comments
-Lines 89-90, the sentence makes no sense.
- Figure 3C, not 3B, should be called line 244. Thus, the authors should switch Figure 2B and Figure 2C.
-Line 264 “transfected cells transfected cells”.
Author Response
Review 2:
Zohar and colleagues evaluated the protective effect of ladostigil on SH-SY5Y cells exposed to Sin1-induced oxidative stress. This manuscript presents interesting data useful for deciphering the molecular effect of ladostigil in neurons. The scientific approach and hypotheses are of interest. Unfortunately, this manuscript is not mature enough yet: the overall organization of the manuscript should be improved, the interpretation of some sets of data is not correct, and the quality of some figures should be improved. Under its current form, this manuscript cannot be published in biomedicines. But provided that the main comments and questions listed below are seriously considered, this manuscript could be resubmitted as it will then be worth being published, in my opinion.
We want to thank the referee for his/her excellent comments that helped us to improve the submission substantially. We improved the presentation, the and the coherent flow of the revised manuscript.
Please note that the numbering in the revised manuscript was changed. The original version included seven figures and three tables. The revised version has six figures and two tables.
Major comments
Title
The title is not accurate as the sets of data presented in this manuscript correspond to the attenuation of the effect of oxidative stress by ladostigil. The effect of ladostigil on ER stress features have not been monitored.
Reply: We have changed the title to better reflect the content: “Ladostigil attenuates Induced oxidative stress in human neuroblast-like SH-SY5Y cells”.
Abstract
In the abstract, the authors stated that oxidative stress induced by Sin1 “resulted in coordinated suppression of ER quality control and ER stress gene sets”. The observation that the expression of genes involved in both ER quality control and ER stress is modified does not preclude these two events are coordinated.
Reply: We rephrase it to emphasize the enrichment of ER genes “resulted in coordinated suppression of endoplasmic reticulum (ER) regulating genes”.
IntroductionThe description of SHSY-5Y cells characteristics made in introduction line 67-69 (ref 31) corresponds to differentiated cells. The authors used undifferentiated cells.
Reply: We added a clarification and mentioned clearly that we have used only undifferentiated setting. We added to the introduction: (Revised lines 74-77) “Undifferentiated SH-SY5Y that contain voltage-dependent calcium channels [32] and active large dense cored vesicles are similar to sympathetic neurons [33]. We used these cells to assess their oxidative vulnerability [34]”.
In the Discussion, we also mention the differentiated cells as potentially model that is likely to be more vulnerable system. Revised lines 432-438: “Upon differentiation of SH-SY5Y (e.g., by retinoic acid, neurotrophic factor), the cells undergo morphological changes, and start expressing major neuronal markers. Differ-entiated SH-SY5Y cells are often used for studying CNS neuron properties (e.g., dopa-minergic neurons, glutamate to GABA conversion, neuronal-like vesicle secretion) [76]. This system might be useful to assess the impact of ladostigil. It was shown that the vulnerability of differentiated SH-SY5Y to stress is enhanced [73,74].”.
Material and methods
-What is used to solubilize ladostigil? (water) ddw
Are untreated cells treated by ladostigil solubilization solution? No need
Please precise what is triggering the fluorescence change of molecular sensors cyto-roGFP and mito-matrix-roGFP. We included a detailed reference that calibrate the GFP-oxidation sensor.
Which SHSY-5Y cell number was used for oxidative stress experiments? If we understand the question, we used 50,000 cells as a lower number, see Supplementary Figure S1.
Which transfection reagent was used? A weird sentence lines 89-90 suggest it may be lipofectamine-2000? Sorry for the careless writing. We fixed this sentence and added reagent source.
Propidium iodide is mentioned in material and methods section but it seems it has not been used in the presented experiments. It was used as mentioned in the FACS description.
PCR extension was really 5 minutes for 300-500 pb amplicons (line 130)? Thanks for noticing this missing information. We skipped by mistake to mention the elongation cycle (30 sec, each cycle) the 5 min was used for final extension. We fixed the writing.
Were real time PCRs performed with the same primers as RT-PCR described in Table 1? No, these are different primers that were added to Revised Table 1, their catalog number and their amplicon length.
Results
-Title of section 3.1 has no meaning. Moreover, as the authors only looked for immunology-related genes, they cannot conclude that an immunological signature dominates in the brain of aged rats.-The authors did not explain why they choose to specifically monitor immunology-related gene expression.No house-keeping gene was used as control in Figure 1.
Reply: The purpose of the original section 3.1 was to connect the reader with the complexity of ‘in vivo’ (i.e., old rat brain). We remove all together this section (as well as original Figure 1) and referred briefly to published results (Linial et al. 2020). We emphasize the rationale for the benefit of a simplified, and controllable in vitro cellular setting.
It would be useful to make sure that gene expression upregulation is a general age-related effect.-There is no Y axis legend in Figure 1.-There is no standard deviation in Figure 1.-Indicate the age of the rats in Figure 1. Adult and old is not precise enough. Young rats (6.5-month-old) and old rats (22-month-old) could be clearer.
Reply: We removed Figure 1 all together,
Which strain of rats was used to generate Figure 1 data? How many animals? Which sex?
Reply: Not relevant, Figure was removed.
It is not clear why the authors stopped using H2O2 and why they used Sin1 to induce oxidative stress instead of classical compounds like paraquat for example?
Reply: H2O2 was shown to act in a short-lived stress. Sin-1 on the other hand is maximal effect in within 5-8 hrs and provide a prolong accumulating stress. Importantly, Sin-1 continues to produce reactive agents within cells. We added information on Sin1 mode of action and references on the important of Sin1 to affect brain mitochondria. Revised lines 210-213: “We replaced H2O2 with Sin1 (3-morpholinosydnonimine), a reagent that promotes a prolong oxidative stress [43] and was shown to damage brain mitochondria [44]. Sin1 is a long-lasting stressor that simultaneously generates nitric oxide (NO) and superoxide (O2−), and act as peroxynitrite donor in cells.”.
Please give more information about Sin1, like its capacity to spontaneously liberate NO when in solution
Reply: Added. Revised lines 210-213:
What is the effect of ladostigil on Sin1-induced oxidative stress when cells are not preincubated with ladostigil?
Reply: We added a sentence that address it. Revised lines 103-105: “A protocol in which ladostigil and the subjected stressors were incubated together failed to monitored a robust biological effect (data not shown). Thus, Ladostigil was added to the culture medium 2-hrs prior to the activation of oxidative stress to cells”.
The treatments’ durations are not clearly indicated on Figure 3. Cells were incubated with ladostigil for 1 or 2 hours?
Sorry for the confusion, we established 2 hrs of pre-incubation.
On figure 3C, Sin1 treatment was 3, 5 and 8 hours as indicated on the bottom of the figure or 3 to 24 hr as indicated in the legend?-
Corrected
If figure 3B, 3C and 3D present data from experiments repeated 3-4 times, the standard deviations values should be indicated, together with statistical analyses.-Figure 3D should show the values of gene expression levels in the absence of Sin1 to see if Sin1 has any impact by itself.
Reply: We improved the presentation of original Figure 3 (now Figure 2). The Figure 2B, 2C (revised) are extracted from FACS assays. Each experiment is based on internal controls to determine the gating and the partition of ~50,000 cells (see Supplementary Figure S1). We included for Figure 2C statistical analysis. We removed Figure 2D to keep the figure more readable and simpler and added a more quantitative data on new Figure 3 that indeed now the values of Sin1 alone, as requested.
The authors did not explain why they choose to specifically monitor Sod1, Sod2 and GPX1 genes expression.
Reply: This set of genes are well known redox / ROS affecting genes. We mentioned that Sod genes catalyze the dismutation of superoxide radical anion to H2O2 and free oxygen. We added a general statement in Revised lines 238-242: ”We questioned whether the observed physiological effect of Sin1 with and without ladostigil is reflected by the expression level of genes controlling cellular redox levels..”.
Figure S3A interpretation is wrong. The authors probably wrongly annotated the wells. Moreover, one cannot draw any conclusion from single-PCR loaded on an agarose gel. This figure should be removed from the manuscript or this set of data should be strengthened. The aim of the manuscript is to study the effect of ladostigil on oxidative-stressed cells.
Reply: We deleted the Figure S3 altogether (sorry for the confusion). We provide a simpler and clear PCR results in Revised Figure 3. In Figure 3A we provide qualitative results (we removed the over analysis from the text). It is evident that with Sin1 and high level of ladostigil, suppression of expression is seen. We added new data in Figure 3B that is based on qPCR (TaqMan) for the effect of Sin1 and Sin1+ ladostigil. This revised figure substantiates the redox sensitivity (including additional quantitation for Mif and the major oxidation transcription factor Nrf2).
Thus Figures 4 and 5 and Table 2, in which the effect of Sin1 alone on gene expression is described, should be linked to figure 7.
Reply: We deleted original Figure 6 that ‘broke’ the flow and the logic of the manuscript.
-How do the authors explain that no apoptosis, ER stress and UPR genes are up or downregulated upon Sin1 and Sin1+ladostigil treatments?
Reply: I assume that we were not clear enough. Actually Figure 7 (=revised Figure 6) ONLY discuss differential gene expression of Sin1 and Sin1+ladostigil treatments. Sin1 led to strong changes in expression (Revised Figure 4-5). Revised Figure 6 shows exactly this. The signal of Sin1 remains solid and only a handful of genes were changed by ladostigil. This is an important observation that shows the rather specific ladostigil effect.
The authors do no explain why they were interested in lncRNA expression.
Reply: Actually, in the original version we explain why lncRNA expression is of interest (exosome, local translation etc). As it is ‘side walk’, we decided to remove it (including original Figure 6 and Table 3).
Moreover, I don’t see the point of showing the effect of Sin1 alone on lncRNA expression in this manuscript. What would have been interesting is the effect of ladostigil+Sin1 on lncRNA expression.
Reply: We removed it all together. But as explained above ladostigil did not ‘cancel’ the induction of lncRNA that was detected following Sin1. As suggested we remove the Figure and the discussion on lncRNA all together.
The sentence line 517 is wrong as ladostigil effect on lncRNA expression level has not been studied here. As suggested we remove the discussion on lncRNA all together. Instead we speculate on the role of ladostigil in view of the observation presented in this study. Revised lines 433-437.
What is the effect of ladostigil itself on genes and lncRNA expression levels?
We remove the discussion on lncRNA all together.
Title of section 3.6 has no meaning.
We remove section 3.6.
-Table S1 and Table S2 are missing in the supplementary file.
We sent them as to the editor in chief. Apparently, it was not available to you. We made sure to re-upload these supplementary results. Moreover, we provide all raw data of the RNA-seq to data shared platform of ExpressArray.
-What is the difference between Figure 7 and Figure S4 besides their formats?
Reply: Supplementary Figure S4 (Revised Figure S3) uses the same data as Figure 7. Figure S4 do not provide a statistical view but instead shows the ‘raw data’ (log CPM / log FC). The reader can appreciate the expression of all genes (including those filtered out due to low expression). Some people may find it useful to look at primary data and not only to see processed results.
Minor comments
-Lines 89-90, the sentence makes no sense.
Sorry, corrected
- Figure 3C, not 3B, should be called line 244. Thus, the authors should switch Figure 2B and Figure 2C.
Both Figures were corrected, revised and simplified
-Line 264 “transfected cells transfected cells”.
Corrected, sorry for the careless repetition.
Round 2
Reviewer 1 Report
Please see attached file. New comments are in RED

Reviewer 2 Report
Review of Manuscript ID: biomedicines-1360915 – Second Round
Ladostigil attenuates oxidation and ER stress in human neuroblast-like SH-SY5Y cells, by Keren Zohar, Elyad Lezmi, Tsiona Eliyahu and Michal Linial.
Zohar and colleagues made some major changes and the manuscript is now more straightforward. However, a large number of poorly written sentences and awkward approximations and shortcuts (genes are not induced or suppressed; their expression is induced or increased; ladostigil is not properly spelled…) make this manuscript still in need of serious proofreading. I’m sorry to tell that it is not a reviewer's job to correct this type of details; a proper proofreading by the authors and particularly by the principal investigator should have corrected these mistakes and avoid the feeling of quickly but badly done. Figure S3 is still not fixed. Under its current form, this manuscript is not ready yet for a publication in biomedicines. But providing a serious and proper proofreading by the authors and the principal investigator and the consideration of a few missing points listed below, this manuscript could be worth being published, in my opinion.
Minor and major comments
Abstract
-Line 18: I still don't agree with the sentence “oxidation by Sin1 resulted in coordinated suppression of endoplasmic reticulum regulating genes…”. The data presented here do not show coordination. The term “simultaneous” would be more appropriate.
Moreover, gene suppression means their deletion. Please replace this sentence by “oxidation by Sin1 resulted in the reduction of the expression level of endoplasmic reticulum regulating genes…”. This type of shortcut is too often used in this manuscript.
-Line 16: “..a decline in cell viability and oxidative levels were partially reversed.”
This sentence means that ladostigil reversed oxidative level reduction? I think you meant to say the contrary. I suggest:….”the decline in cell viability and the increase of oxidative levels were partially reversed.”
-Lines 20, 25, 51, 181, 191, 230, Figure S2 legend (twice), Figure S3: ladostigil not ladogtigil….
-Line 22: “ …and Synj1 22 (Synaptojanin 1) that function in ER-autophagy and endocytic pathways. …”. Replace “function” by “are involved”.
-Line 24: “Therefore, under conditions of chronic stress that signify aging brain, …”. Replace “signify” by “are observed in”.
Introduction
-Line 36: Change “A phenomenon cell characteristic of the aging brain is a reduced capacity to cope with oxidative stress compared to cells from young and healthy adult brains.” by “A characteristic phenomenon of the aging brain cells is a reduced capacity to cope with oxidative stress compared to cells from young and healthy adult brains.”
-Line 73: delete “simplified” as it isn’t clear compared to what SH-SY5Y are simpler.
Results
-On Figure 2A, ladogtigil seems to increase cell proliferation in the absence of Sin1. Moreover, Sin1 treatment also seems to increase cell proliferation at 50 and 100 µM. Please comment.
-As requested in the 1st round of review, the standard deviations values should be indicated for Figure 2B, together with statistical analyses.
-Line 146: “Table 1. RT-PCR on selected genes that participate in the cell redox balance.”.
-Line 147: “a The two expected sizes of the PCR products resulting from alternatively spliced variants.”
-Line 184 replace “we tested the neuroblastoma…” by “we used the neuroblastoma…”.
-Line 185 replace “in testing the sensitivity…” by “in order to test the sensitivity…”.
-Line 188 replace “…can dissipate moderate levels of H2O2…” by “..can dissipate moderate levels of H2O2-induced oxidative stress”.
-Line 199 “..To test a possible shift in the cellular redox state in the presence of H2O2, we utilized molecular sensors for oxidation…”.
-Lines 200-201: “Cells were transfected with a mixture of cytosolic and mitochondrial fluorescence sensors for monitoring their redox state.”
-Lines 205-207: The effect 205 of ladostigil on the cell oxidative state was undetected 24 hrs after exposure even at high concentration (Supplemental Figure S2). “. I guess it is Figure S1 that should be called, not S2?
-Lines 211-212: “We replaced H2O2 with Sin1 (3-morpholinosydnonimine), a reagent that promotes a prolonged oxidative stress [43] …”.
-Lines 227-228: “A maximal effect was measured 5 hrs following the insult.”. Precise it was with 300 µM Sin1.
-Lines 243-244: “the expression of the gene encoding Sod2, a mitochondrial resident enzyme, and Gpx1 was induced from their basal level by Sin1.”
-Lines 246-249: “Ladostigil reduced the expression of the genes encoding the antioxidant superoxide dismutase enzymes Sod1, Sod2, Gpx1 and Nfe2l2, a major oxidation sensitive gene regulatory transcription factor (Nrf2) reduced their expression by ~50-60% of the maximal induced level by Sin1.”
-Lines 250-251: “Similarly, in the presence of high concentration of ladostigil, and Sin1 induction Mif expression level was 39% of its level induced by Sin1 in the presence of Sin1.”
-Lines 274-275: “(A) Volcano plot of the differentially expressed genes that were induced (red) and suppressed (blue) according to the fold change and statistical significance. “
-Line 265: “3.3. A coordinated simultaneous suppression of ER chaperones and folding enzymes by a chronic oxidative stress”.
-Lines 284-285: “… and up-regulated (Figure 4C) genes following Sin1 treatment relative to non-treated cells and assessed their annotations' enrichment according to GO (Gene Ontology) …”.
-Lines 289-290: “…with dominant GO biological process terms concerning neuronal development and differentiation.”
-Lines 295-297: “Figure 5. A collection of genes which expression is significantly suppressed reduced genes (>2 folds) following exposure to Sin1 with respect compared to untreated cells (24 hrs). Expression levels are in a logarithmic scale. Genes are sorted 296 alphabetically.
-Line 302: “Dnajc3, Xbp1, Hspa5, c, Hyou1, Calr, Hspa1b, …”. What is gene c?
Hsp90b1 is not shown on Figure 5?
-Line 322: “Table 2 lists the genes which expression is significantly induced genes (>2-folds).”
-Lines 325-326: “….is among the few highly expressed induced genes.”
-Lines 337-338: “an ATP receptor that following its which activation, leads to change in Ca2+ homeostasis…”.
-Lines 342-343: “Incubation of SH-SY5Y cells with ladostigil under basal conditions had a negligible effect on basal gene expression.”.
-Lines 344-345: “…relative to untreated cells identified only 7 genes which expression levels statistically significantly changed genes (total 11,588 genes…”.
-Lines 352-353: “However, it the nature of affected genes suggest involvement of the 352 mitochondria and ER.”.
-Lines 354-356: “Figure 6 shows the differential expression of ladostigil on genes expression in cells exposed to Sin1 and pretreated with ladostigil. Altogether, only 10 genes (TMM >5; p-value FDR <0.05) resulted in show a significant differential expression…”.
-Lines 372-373: “Synaptojanin 1 (Synj1, p-value FDR = 2.1E-05) is among the very few genes downregulated genes by ladostigil.”.-Lines 396-397: “and chronic diseases that are signified characterized by a strong interplay between oxidative and ER stress…”.-Lines 416-417: “Down-regulation of genes involved in ER, chaperons, and quality control signifies is a characteristic the response of Sin1-exposed cells (50 µM, 24 hrs).”
-Figure S3 legend is still wrong at several levels, as I already mentioned in the first round reviewing process:
…” while Sod2, a mitochondrial resident enzyme, was induced >2-fold from its basal level.”
Sod2 basal level is not detectable; thus, it is not possible to say that the expression level of this gene increased by >2 folds.
… “Ladostigil at 5.4 μM (but not 54 μM), reduced Sod2 to its basal level.”.
With 5.4 µM ladostigil, I can see an increased intensity of Sod2 PCR band, not a decrease. An increased intensity of Sod2 PCR band is also visible with 54 µM ladostigil.
….”A similar pattern was recorded for Glutathione peroxidase 1 (Gpx1), a ubiquitous enzyme that is located in the cytosol, mitochondria, and peroxisomes.”
With 5.4 and 54 µM ladostigil, I can see an decreased intensity of Gpx1 PCR band which basal expression level is not detectable in the absence of Sin1 or ladostigil.
I’m thus still concerned by the data presented in Figure S3 and by the interpretation the authors made of the presented data. I suggest this Figure S3 should be removed from the revised version; but it can definitively not be left like it is now.
-Figures S2, S3 and S4 are not called in the manuscript.
Material and methods
-Please precise what is triggering the fluorescence change of molecular sensors cyto-roGFP and mito-matrix-roGFP, respectively.I wasn’t clear enough: please precise which cellular changes do you measure using these probes?-Which SHSY-5Y cell number was used for oxidative stress experiments?Precisely in experiments shown in Figure 1A and Figure 2.
